# The Estrogen Receptor α Signaling Pathway Controls Alternative Splicing in the Absence of Ligands in Breast Cancer Cells

**DOI:** 10.3390/cancers13246261

**Published:** 2021-12-13

**Authors:** Jamal Elhasnaoui, Giulio Ferrero, Valentina Miano, Santina Cutrupi, Michele De Bortoli

**Affiliations:** 1Department of Clinical and Biological Sciences, University of Turin, Orbassano, 10043 Turin, Italy; jamal.elhasnaoui@unito.it (J.E.); giulio.ferrero@unito.it (G.F.); santina.cutrupi@unito.it (S.C.); 2Department of Computer Science, University of Turin, 10149 Turin, Italy; 3Department of Pathology, Division of Cellular and Molecular Pathology, Addenbrooke’s Hospital, University of Cambridge, Cambridge CB2 0QQ, UK; vm403@cam.ac.uk; 4Center for Molecular Systems Biology, University of Turin, 10124 Turin, Italy

**Keywords:** breast cancer, estrogen receptor, alternative splicing, EMT, splicing signature

## Abstract

**Simple Summary:**

Aberrant alternative splicing is now considered a hallmark of cancer, including breast cancer. This results in the production of novel tumor-specific splice RNA variants, and the activation of biological processes such as epithelial-to-mesenchymal transition, leading to more aggressive phenotypes. The purpose of this study was to explore the role of estrogen receptor α in regulating the expression of RNA-binding proteins in luminal breast cancer cells and to determine the effects of its downregulation at the isoform level by exploring changes in isoform usage and alternative splicing. The findings of this study unravel a novel layer of gene regulation mediated by estrogen receptor α, which is fundamental for breast cancer cell growth as well as epithelial-to-mesenchymal transition. Finally, we foresee that this novel feature should be considered when studying the functional roles of estrogen receptor α in the onset and progression of breast cancer.

**Abstract:**

Background: The transcriptional activity of estrogen receptor α (ERα) in breast cancer (BC) is extensively characterized. Our group has previously shown that ERα controls the expression of a number of genes in its unliganded form (apoERα), among which a large group of RNA-binding proteins (RBPs) encode genes, suggesting its role in the control of co- and post-transcriptional events. Methods: apoERα-mediated RNA processing events were characterized by the analysis of transcript usage and alternative splicing changes in an RNA-sequencing dataset from MCF-7 cells after siRNA-induced ERα downregulation. Results: ApoERα depletion induced an expression change of 681 RBPs, including 84 splicing factors involved in translation, ribonucleoprotein complex assembly, and 3′end processing. ApoERα depletion results in 758 isoform switching events with effects on 3′end length and the splicing of alternative cassette exons. The functional enrichment of these events shows that post-transcriptional regulation is part of the mechanisms by which apoERα controls epithelial-to-mesenchymal transition and BC cell proliferation. In primary BCs, the inclusion levels of the experimentally identified alternatively spliced exons are associated with overall and disease-free survival. Conclusion: Our data supports the role of apoERα in maintaining the luminal phenotype of BC cells by extensively regulating gene expression at the alternative splicing level.

## 1. Introduction

Alternative splicing (AS) is a complex regulatory mechanism of gene expression which is dysregulated in many oncological contexts, including a wide range of cancers [1]. Indeed, AS dysfunction is now considered a new hallmark of cancer, as this alteration has an impact on the splicing patterns of different oncogenes and tumor suppressor genes, including transcription factors (TFs), splicing factors (SFs) and RNA-binding proteins (RBPs) [2,3]. One such cancer type is breast cancer (BC), in which AS dysregulation is one of the main steps involved in the development and progression of the disease [4,5]. Ranking number one in women’s cancer-related deaths worldwide, BC is a heterogeneous disease covering four different subtypes characterized by distinct molecular and clinical phenotypes, among which the luminal estrogen receptor α positive (Erα+) subtype [6] is the most frequent, representing up to 80% of diagnosed cases [7]. The ERα+ BC subtype is clinically characterized by being mildly aggressive and by showing an optimal response to targeted endocrine therapies [8].

ERα, together with other transcription factors such as FoxA1, TFAP2C, and GATA3 are key factors in the determination and maintenance of the epithelial phenotype of mammary cells, as activated not only by estrogen, but also by other signaling pathways [9,10]. This activity is reflected in BC, since tumors that retain the expression of ERα show several epithelial features and, clinically, are less invasive and aggressive than other subtypes exhibiting mesenchymal features [11]. The transcriptional activity of ERα in breast tumor cells has been the subject of an impressive number of research papers in the last decades, especially because it represents one of the prototypes of druggable molecules in cancer, testified by the success of Tamoxifen and other antiestrogens in BC treatment since 1975 [12]. Genome-wide studies have shown the relevance and wideness of the ERα-dependent transcriptional response following the stimulation of cultured BC cells with either estrogen or anti-estrogenic compounds, but also in its unliganded (apoERα) form [13,14].

Moreover, several recent studies have shown that the action of ERα is not limited to controlling the transcription of protein-coding genes [15], but also actions such as controlling the noncoding elements comprising of enhancer RNAs (eRNAs) [16,17], long-noncoding RNAs (ncRNAs) and microRNAs (miRNAs), delineating a more complex regulatory network which includes post-transcriptional regulation [14]. Other groups have demonstrated that ERα coordinates its transcriptional output with the selective modulation of the mRNA translation process [18]. Very recently, an intriguing study by Xu and colleagues reported that ERα directly binds several RNAs through its hinge domain, resulting in AS and translational control of target RNAs [19]. These findings highlight novel ERα features in controlling several aspects of RNA biology in BC.

In our lab we addressed whether the hormone-independent activity of apoERα in tumor cells regulates gene transcription as demonstrated in other experimental model systems [20]. Indeed, apoERα is needed to maintain an active, euchromatic status of the E-cadherin coding gene [21], an essential protein for preserving the epithelial cell phenotype. At a genome-wide level, the transient downregulation of apoERα in MCF-7 BC cells allowed a description of more than 4000 apoERα binding sites, regulating the transcription of genes related to cell proliferation and epithelial differentiation [13]. While these genes were within the larger group of estrogen-regulated genes, ontology terms exquisitely related to the control of the epithelial phenotype emerged. Among these genes, the RBP class was outstanding (more than 680, of which 85 represented SFs), thus suggesting that ERα may control gene expression at different levels, such as the regulation of AS.

In this work, we took advantage of the RNA-seq data of apoERα-targeted MCF-7 cells to explore the transcriptomic changes in terms of transcript isoform usage and AS. Results demonstrated that even in an absence of estrogen stimulation, ERα exerted an extensive regulation of gene expression at a level further than transcription. Comparisons with tumor-derived data support the relevance of the activity of ERα in primary luminal BCs.

## 2. Results

### 2.1. apoERα Activity Regulates the Expression of RBPs and SFs in the MCF-7 BC Cell Line

To explore the functional role of apoERα activity on the gene expression process in BC, a differential gene expression (dGE) analysis was performed on our previously published paired-end RNA-seq experiment [22] consisting of hormone-deprived MCF-7 BC cells treated with a control, or with *ESR1*-targeting siRNA (siCTRL vs. siERα). Silencing apoERα in MCF-7 significantly perturbed the expression of 6611 genes (|log2FC| > 0.2 and adj. *p* < 0.05), where 3741 were downregulated and 3140 were upregulated (Appendix A). As expected, the functional enrichment analysis showed the downregulated genes as enriched in terms of cell cycle progression, including cell proliferation, DNA replication, and DNA damage repair (Appendix A), in line with the cellular phenotype previously reported [13]. Conversely, upregulated genes were particularly enriched in EMT-related processes, including actin cytoskeleton organization, cell movement, cell morphogenesis involved in differentiation, developmental growth, and the positive regulation of cell migration (Appendix A).

Interestingly, apoERα depletion induced significant expression changes to 681 RBPs, most of which were downregulated (486, 71%) (Figure 1a,b and Appendix A). A total of 84 RBPs (12.5%) were represented by SFs, of which 63 (75%) and 21 (25%) were down- and upregulated, respectively (Appendix A). The functional enrichment analysis indicated that downregulated RBPs were enriched in terms of RNA processing, including translation, ribonucleoprotein complex assembly and biogenesis, RNA localization, and 3′end processing (Figure 1b and Appendix A). Conversely, upregulated RBPs were exclusively enriched in other processes, such as the ncRNA metabolic process, actin cytoskeleton filament organization, the regulation of binding, and the serine/threonine kinase signaling pathway (Figure 1c and Appendix A).

The depletion of apoERα causes a significant decrease in the expression of epithelial-specific SFs, such as *ESRP1* and *ESRP2*, the core splicing regulatory proteins in epithelial cells [23,24], while inducing the expression of EMT-related RBPs such as *QKI* [25] and *SMAD4* [26] (Appendix A). Therefore, to further explore the link between the apoERα-regulated RBPs and the EMT process, the apoERα-regulated RBPs were overlapped with two independent public lists of RBPs. The first list was identified as DE between epithelial and mesenchymal BC cell lines [23] (Appendix A), and the second list included DE RBPs upon the induction of the EMT process by the overexpression of the EMT activator *ZEB1* in the H358 epithelial cells [27] (Appendix A). This analysis revealed 54 RBPs as DE in all the three datasets (Figure 1d and Appendix A). Interestingly, the overlap with both studies indicated that apoERα depletion induced RBPs that were highly expressed in mesenchymal cells, whereas it decreased the expression of RBPs that were highly expressed in epithelial cells (Figure 1e and Appendix A). Furthermore, to search for the possible associations between ERα and these apoERα-regulated RBPs, a correlation analysis between the expression of ERα and of these RBPs was performed considering RNA-seq data from 772 ERα+ BC samples from TCGA [28]. This analysis revealed two sets of RBPs comprising of 230 (55%) DE RBPs that were positively correlated (rho > 0.10, *p* < 0.005), and 188 (45%) RBPs that were negatively correlated (rho < −0.10, *p* < 0.005) with ERα expression (Appendix A). In particular, 168 RBPs (out of 230, 73%) that were correlated with ERα showed a significant downregulation following apoERα depletion. Similarly, 94 RBPs (out of 188, 50%) that anti-correlated with ERα were upregulated in our dataset (Figure 1f and Appendix A). Importantly, a hallmark gene set enrichment analysis showed that anti-correlated RBPs were enriched in terms of apical junction organization, EMT, hypoxia, and P53 signaling pathway hallmarks (Figure 1g and Appendix A), whereas the positively correlated RBPs were enriched in terms of the cell cycle and proliferations, such as Myc targets, E2F targets, G2M checkpoint hallmarks, and the stress-related response such as unfolded protein response hallmarks (Figure 1h and Appendix A). Selected examples representing the top three EMT-related RBPs, correlating (MSI2, *ESRP1*, and FKBP4) and anti-correlating (*SAMD4A*, *QKI,* and MBNL1) with ERα expression, are shown in Figure 1i.

Moreover, the comparison of apoERα-regulated RBP genes to a previously published list of RBPs reported as DE between ERα+ BC tissues and their adjacent normal counterparts [29] revealed 413 RBPs common to both datasets (exact hypergeometric probability, *p* < 0.0001) (Appendix A). The overlapping DE RBP genes included 130 (31%) RBPs coherently regulated in both datasets (83 downregulated and 47 upregulated), and 283 (69%) RBP genes showed an opposite expression change (233 downregulated in apoERα silencing while upregulated in tumors, and 50 upregulated in apoERα silencing while downregulated in tumors) (Appendix A).

### 2.2. EMT-Related Gene Isoforms Are Differentially Expressed upon apoERα Silencing

To identify genes with isoform switching events driven by apoERα depletion, a differential isoform usage (dIU) analysis was performed using IsoformSwitchAnalyzeR [30] (Appendix A). This analysis revealed 605 genes with isoforms that were differently regulated upon apoERα depletion, with a total of 758 isoforms involved in significant switching events (adj. *p* < 0.05 and |dIF| > 0.05) (Figure 2b,c and Appendix A). The functional enrichment analysis of genes harboring significant switching isoform pairs revealed terms related to cell cycle regulation (e.g., mitotic cell cycle phase transition) and to cell migration (e.g., actin filament organization, cell-cell junction organization, and the establishment of vesicle localization) (Appendix A). The annotation of switching isoforms showed that apoERα depletion induced an enrichment of isoforms characterized by specific structural changes (Figure 2d and Appendix A). Specifically, apoERα silencing increased the expression of isoforms characterized by longer 3′UTRs (adj. *p* < 0.001), longer 5′UTRs (adj. *p* < 0.01), more protein domains (adj. *p* < 0.001), more intron retention (IR) events (adj. *p* < 0.01), an insensitivity to nonsense-mediated decay (NMD) (adj. *p* < 0.01), and by having more coding than non-coding transcripts (adj. *p* < 0.01) (Appendix A). Again, genes harboring isoform switching events with putative downstream structural consequences were enriched in terms of the cell cycle progression and DNA repair (e.g., the DNA damage checkpoint, the G2 DNA damage checkpoint, and the regulation of chromosome organization) and cell migration (e.g., actin filament-based process) (Appendix A).

The comparison of the induced and repressed isoforms indicated an enrichment of putative AS events (ASEs) possibly underlying the observed isoform switching events, including exon skipping (ES) and IR events (adj. *p* < 0.05), differential transcription start site (TSS) usage (adj. *p* < 0.05), and differential transcription termination site (TTS) usage (adj. *p* < 0.0001) (Figure 2e and Appendix A). As shown in Figure 2f, *USO1* is a clear example of a gene with switching isoform pairs identified in our analysis. In this gene, the apoERα depletion resulted in a significant switch in the relative abundance of the two isoforms (ENST00000514213.6 induced, and ENST00000264904.8 repressed) characterized by the differential inclusion of two alternative exons, of which one encodes for a domain (*PF04869*) involved in the dimerization of the globular head of the protein (Figure 2g).

Interestingly, the dIU analysis revealed peculiar cases in which the gene- and isoform-specific responses to apoERα depletion were non-concordant. For instance, the expression analysis showed 120 isoforms to be repressed, while their parent genes were upregulated by the dGE analysis (Appendix A). Similarly, apoERα depletion appeared to induce 160 isoforms, while their parent genes were downregulated by dGE. On the other hand, several genes were not regulated by the dGE analysis but were significantly regulated at the isoform level only, as exemplified in Appendix A.

### 2.3. apoERα Depletion Induces Internal ASEs in EMT-Related Genes

To further investigate the ASEs regulated by apoERα depletion, a differential AS analysis was performed. The analysis revealed 825 ASEs upon apoERα depletion, of which 546 (65%) classified as ES, followed by 145 (17%) that were classified as IR, 73 (9%) that were classified as A3′, 45 (5%) that were classified as A5′, and 37 (4%) mutually exclusive exon (MX) events (Figure 3a,b and Appendix A). The differential inclusion levels (dPSI, differential percent spliced-in index) for the significant ASEs identified is reported in Figure 3c. The dPSI of most of the significant ASEs falls within the range of −0.2 to 0.2, except for RI events, where the dPSI of most of the events falls within the range of −0.1 to 0.1.

The top 100 significant ASEs were ES events and the top 50 of these are shown in Figure 3d. Upon apoERα depletion, most of ASEs (70%) had a dPSI greater than 0 (Figure 3d and Appendix A). Importantly, in line with the dIU analysis, the functional enrichment analysis of genes harboring ASEs showed an enrichment in terms of EMT, such as actin cytoskeleton organization, actin filament-based movements, vesicle mediated transport, as well as cell progression processes such as mitotic cell cycle phase transition, DNA mismatch repair, spindle organization, the chromosome segregation process, and the mRNA metabolic process; and metabolism processes including the phospholipid metabolic process, the hexose metabolic process, and the carbohydrate derivative biosynthetic process (Figure 3e and Appendix A). Selected examples of EMT-related genes harboring the most significant ES events are reported in Figure 3f.

To explore our hypothesis that apoERα controls EMT-related ASEs, the apoERα-modulated ASEs were compared with 191 ASEs that were reported as significantly differentially regulated between epithelial and mesenchymal BC cell lines by Shapiro and colleagues [23]. Interestingly, 26 AS genes overlapped between the two datasets (exact hypergeometric test, adj. *p* < 0.0001) (Appendix A). In particular, 26 ES and 2 MX events were common between the two datasets, of which 19 ES events and 1 MXE event were coherently regulated. For example, the most significant ES event upon apoERα silencing was an ES of the 7th exon of the amyloid beta precursor-like protein 2 (*APLP2*) gene (dPSI = −0.176; adj. *p* < 0.0001) ranked as the top third significant event in the study of Shapiro et al. Similarly, the ES event (dPSI = −0.25; adj. *p* < 0.0001) in the *USO1* vesicle transport factor (*USO1*) gene was also repressed in this dataset. Furthermore, the set of apoERα-modulated ASEs overlapped with a published list of ASEs occurring in a 7-day time-course experiment of EMT upon the overexpression of the EMT activator, *ZEB1*, in the H358 epithelial cells [27]. This analysis revealed 105 (60 ES, 10 MX, 27 IR, 5 A3, and 3 A5′) overlapping ASEs. Of this, 75 (71%) ASEs were coherently regulated (Appendix A).

### 2.4. The apoERα-Regulated RBPs Are Significantly Correlated with ERα mRNA Levels in ERα+ BCs and Are Predicted Regulators of the apoERα-Modulated ASEs

To determine candidate SFs potentially regulating the identified ASEs in our dataset, a differential RBP-binding motif enrichment analysis was performed for the regions involved in each ASE type, based on the direction of regulation (i.e., dPSI > 0.05, dPSI < −0.05). This analysis revealed a total of 95 enriched SF-binding motifs (37 enriched for the ES events, 41 for IR events, 49 for A5′ events, 61 for A3′ events, and 91 SFs enriched for MXE events) (Appendix A). Importantly, while showing a preferential binding depending on the direction of regulation and the ASE type, most of the enriched SFs were common among ASE types (Appendix A). Conversely, the binding motifs of 17 and three SFs (TUT1, TIA1, and TIAL1) were exclusively enriched in MX and A3′ events, respectively. In the case of ES events, the enriched SF motifs were prevalently predicted upstream of the spliced exons, whereas in the case of EI events, the enriched SF motifs were prevalently predicted within the spliced exons (Appendix A). Among the 95 enriched SFs, 49 were DE upon ERα silencing, either at the gene or isoform level and most of them (85%) were significantly downregulated (Appendix A). In addition, 57 enriched SFs exhibited either significant AS changes or were significantly correlated with ERα in primary BCs (Appendix A). A group formed by 11 SFs that were enriched in ASEs correlated with ERα mRNA levels in primary tumors and exhibited significant regulations both at gene and AS levels upon apoERα depletion in MCF-7 cells. The top three significant SFs include *SAMD4A* (log2FC = 0.91; adj-*p* < 0.0001; rho = −0.45; rho *p* < 0.0001), *CELF1* (log2FC (*CELF1*) = −0.13; adj-*p* = 0.11; log2C (CELF1-204) = −0.52, adj-*p* (CELF1-204) < 0.0001; log2FC (CELF1-202) = 3.47, adj-*p* (CELF1-202) < 0.0001; rho = 0.20; rho *p* < 0.0001) and *QKI* (log2FC = 0.27; adj-*p* < 0.01; rho = −0.38; rho *p* < 0.0001). Although they were not considered for the RBP-binding motif enrichment analysis because of their missing position weight matrices and consensus binding motifs, a second group of 73 SFs were correlated with ERα mRNA levels in primary tumors and also exhibited significant AS changes upon apoERα depletion (Appendix A). Noticeably, hnRNPL, which is known to interact with DSCAM-AS1 [31,32], an apoERα-regulated lncRNA [33], was regulated at the isoform level and was positively correlated with ERα mRNA levels in primary tumors (log2FC (HNRNPL-210) = −0.37; adj-*p* < 0.05; rho = 0.23; rho *p* < 0.0001).

### 2.5. apoERα-Regulated Exons Are Differentially Included in Primary BCs and Correlate with ERα mRNA Levels

The identified apoERα-modulated ASEs were further explored in 965 BC samples including 773 ERα+ BCs, 192 ERα- BCs, and 113 adjacent normal samples (Appendix A) using the data from TCGASpliceSeq database [34]. Among the apoERα-regulated ASEs, 228 (28%) were detected in these data (Appendix A). Interestingly, among them, 81 ASEs were significantly correlated (*p* < 0.05) with Erα mRNA levels in Erα+ BC samples (50 and 38 positively and negatively correlated, respectively) (Figure 4a and Appendix A). The most significantly correlated ASEs (*p* < 0.0001) were in an ES event in the calsyntenin-1 gene (*CLSTN1*) (rho = −0.44; dPSI = −0.05), an ES event in the erythrocyte membrane protein band 4.1-like 1 (*EPB42L1*) gene (rho = −0.40; dPSI = −0.118), an ES event in the myoferlin (*MYOF*) gene (rho = −0.36; dPSI = 0.153), and an ES event in the Ral GEF withPH domain and SH3 binding motif 2 (*RALGPS2*) gene (rho = 0.25; dPSI = −0.103).

The analysis of the inclusion levels of these ASEs in different groups of BC patients showed 140 ASEs characterized by significantly different inclusion/exclusion levels (|dPSI| > 0.05; *p* < 0.05). Specifically, 53 ASEs were differentially included between ERα+ and ERα- BC samples, 72 between BC and normal samples, and 15 between ERα+ BC samples characterized by high or low ERα mRNA levels (Figure 4b and Appendix A). The inclusion/exclusion levels of eight ASEs were significantly different in all the three comparisons (Figure 4b and Appendix A) and all these ASEs were significantly correlated (*p* < 0.05) with ERα mRNA levels in ERα+ BCs (Appendix A). In ERα+ BCs, compared to normal samples, the sixth exon of the collagen type VI alpha 3 chain coding gene (*COL6A3*) involved in an ES event (COL6A3_5_6_7_ES) exhibited a significantly higher inclusion level in tumor samples (dPSI = 0.49; *p* < 0.0001). Another significantly different ES event was observed in the same gene (COL6A3_2_4_5_ES) and involved the third and fourth exons of the *COL6A3* gene. The event had a higher inclusion in tumors as compared to normal samples (dPSI = 0.21; *p* < 0.0001). Selected examples of these ASEs are reported in Figure 4c, and the full list is provided in Appendix A.

The association between the apoERα-modulated ASEs and molecular pathways in ERα+ BCs was then investigated using PEGASAS (details in Section 4) [35]. This analysis revealed two main clusters of molecular pathways characterized by a significant correlation with the inclusion/exclusion levels of the analyzed ASEs (Figure 4d). Specifically, a cluster of 12 ASEs was correlated mainly with EMT-related pathways, including the TGFB_SIGNALING_PATHWAY, EMT, and APICAL_JUNCTION pathways, but also the KRAS_SIGNALING_UP, ESTROGEN_RESPONSE, and CHOLESTEROL_HOMEOSTASIS pathways. The second cluster was related to cell cycle progression related pathways such as DNA_REPAIR, E2F_TARGETS, G2M_CHECKPOINT, MYC_TARGETS_V1, and MITOTIC_SPINDLE pathways, in addition to other metabolism-related pathways such as OXIDATIVE_PHOSPHORYLATION and GLYCOLYSIS. Seven ES events (PLOD2_14_15_16_ES, MYOF_16_17_18_ES, EPB41L1_19_20_21_ES, LMO7_9_12_13_ES, MLLT4_15_16_17_ES, ARGEF11_38_39_40_ES, and CLTSTN1_10_11_12_ES) were correlated (r > 0.3) with terms belonging to the first cluster, whereas most them were negatively correlated with those of the second one (Figure 4d). Noticeably, these ASEs exhibited a differential inclusion/exclusion level between tumor samples and normal samples and were significantly correlated with ERα mRNA levels. In addition, two ASEs (DNM2_14_15_17_ES and SPTAN1_22_23_24_ES) were negatively correlated with all terms of the first cluster and showed a positive correlation with four cell cycle-related terms (E2F_TARGETS, G2M CHECKPOINT, MYC_TARGET_V1, and MYC_TARGET_V2) (Figure 4d and Appendix A).

Furthermore, a survival analysis of ERα+ BC patients from TCGA was performed by stratifying patients based on the inclusion levels of apoERα-modulated ASEs. Twelve ASEs (11 ES events and 1 A3′ event) were significantly associated with patient overall survival (OS) (Appendix A). In particular, a higher inclusion of the 6th exon of *COL6A3* was significantly associated with a longer OS (*p* < 0.01) (Figure 4e and Appendix A). Similarly, a higher inclusion of the remaining 11 ASE events was also associated with a longer patient OS. Noticeably, two ES events in the *COL6A3* gene (COL6A3_2_4_5_ES, COL6A3_5_6_7_ES) were associated with a longer patient OS. On the other hand, 7 ASEs (6 ES events and 1 A3′ event) were significantly associated with patients’ disease-free survival (DFS) (Appendix A). Four and three ASEs were associated with a better and worse DFS, respectively. Notably, a higher inclusion level of the 12th exon of the protein phosphatase 4 regulatory subunit 3B (*PPP4R3B/SMEK2*) gene was associated with both better DFS (*p* < 0.01) as well as longer OS (*p* < 0.05) (Figure 4f and Appendix A).

## 3. Discussion

In the present study, hormone-independent ERα (apoERα) activity was explored in MCF-7 cells at both transcriptional and splicing levels. We demonstrated that apoERα regulates a set of relevant ASEs in tumor tissues which correlate with ERα mRNA levels and show a prognostic value in BC patients. The role of apoERα in maintaining the expression of epithelial genes and in promoting cell cycle progression in MCF-7 cells was evidenced through the analysis of a deep paired-end RNA sequencing experiment. Clearly, apoERα silencing significantly hampered the expression of cell cycle-related genes which are essential for cell proliferation and survival, whereas promoting the increased expressions and activities of a number of mesenchymal markers may result in a more mesenchymal-like phenotype in surviving cells, as previously observed [36]. Interestingly, apoERα silencing significantly repressed the expression of RBP and SF genes which paralleled their significant regulation at the AS level.

The impact of apoERα depletion at the isoform level was evaluated by three independent analyses including the dIE, dIU, and differential AS analyses. Our computational pipeline sheds light on the importance of considering the analysis at the level of isoforms, rather than limiting the attention on the gene level as previously reported [37]. Our data suggest that complex mechanisms at the level of RNA transcripts drive the expression of specific protein isoforms, which may be functionally different. An analysis at the gene level confirmed that Erα, in the absence of hormones, is crucial for cell proliferation and for maintaining an epithelial-like luminal phenotype of MCF-7 BC cells [13,32]. Notably, several cell cycle-related genes such as the E2F transcription factor 1 (*E2F1*) gene, the checkpoint kinases 1 (*CHEK1*) and 2 (*CHEK2*) genes, cyclin dependent kinases 1 (*CDK1*), 2 (*CDK2*), 4 (*CDK4*), 6 (*CDK6*), and 7 (*CDK7*), and minichromosome maintenance complexes 3 (*MCM3*), 4 (*MCM4*), 5 (*MCM5*), 6 (*MCM6*), 7 (*MCM7*) and 10 (*MCM10*) were significantly downregulated by apoERα depletion. Noticeably, all of the aforementioned genes were induced under the 17β-estradiol stimulation of MCF-7 cells [38,39]. On the other hand, genes known to be involved in EMT processes, such as the tumor growth factors beta 1 (*TGFB1*), 2 (*TGFB2*), and 3 (*TGFB3*); their receptors, type 1 (*TGFBR1*), 2 (*TGFBR2*) and 3 (*TGFBR3*); the CD44 antigen (*CD44*) gene; the collagen type V alpha 1 chain (*COL5A1*) gene; the type VI alpha 1 (COL6A1) and 2 chains (*COL6A2*); and the filamin A (*FLNA*) gene were significantly induced by apoERα silencing [40]. Such a transition from the epithelial-to-mesenchymal phenotype was indeed reported in a study [36] showing that by stably knocking down ERα, MCF-7 cells underwent a potent clonal EMT, as well as changes in the expression and activity of matrix macromolecules, finally resulting in BC cell migration and invasion.

Our attention was particularly attracted to the high number of RBP and SF genes regulated by apoERα. An important number of studies compared the transcriptome of human BC versus healthy matched tissues, finding approximately 50% of altered transcripts of genes encoding RBP and SF proteins [41,42]. The EMT splicing signature contains RBP and SF proteins as main components involved in the changing of splicing patterns in in vitro models [23]. Importantly, these changes in splicing patterns were correlated to the concentration level and activity of specific SF genes, especially in cancer [5]. For instance, a previously published work reported a significant association between SF genes and their ER status in different BC subtypes, and their correlation with clinical phenotypes, such as tumor aggressiveness, metastasis, and survival was investigated [43]. The overlap with this study showed that the expression of 76 SFs is also modulated by apoERα (Appendix A). In particular, apoERα silencing repressed the expression of epithelial-specific RBP genes, including the PHD finger protein 5A (*PHF5A*), previously identified as an oncogene frequently upregulated and associated with poor survival in BC [44]. Knocking down this gene significantly suppressed cell proliferation and increased apoptotic signalling by promoting the expression of a short, truncated Fas-activated serine/threonine kinase isoform, enabling Fas-mediated apoptosis in BC cells [44]. Another apoERα-modulated RBP gene is the nucleolar-related dyskerin pseudouridine synthase 1 (*DKC1*) gene, reported as a prognostic marker in BC patients that is associated with poor patient outcomes [45]. *DKC1* overexpression conferred a more aggressive phenotype and increased intrinsic ribosomal activity in cells derived from normal breast epithelium [46].

Furthermore, apoERα depletion caused a significant downregulation of several SF genes related to the epithelial phenotype. For example, *ESRP1* and *ESRP2* genes are the core regulators of AS in epithelial cells and play a crucial role during EMT [23,24,47]. Other examples of downregulated SF genes are the muscleblind-like 1 (*MBNL1*) gene that acts as a tumor suppressor in BC [48] by controlling AS, translation, and RNA decay through binding at 3′UTRs [49,50]. Similarly, apoERα silencing decreased the expression of the *MBNL3* gene, which is downregulated during the EMT of epithelial BC cells [23,51]. The expression of the transformer 2 beta homolog (*TRA2B*) gene, an oncogene in BC acting on the splicing pattern of the *CD44* gene involved in EMT [52], also decreased in our data. On the other hand, apoERα downregulation corresponds with an increased expression of 30 SF genes that were previously reported to be expressed at a higher level in the mesenchymal phenotype in BC [23]. This includes the cytoplasmic polyadenylation element binding proteins 1 (*CPEB1*) gene (the top significant DE SF gene in our dataset), 2 (*CPEB2*), and 4 (*CPEB4*); and the eukaryotic translation initiation factor 4E family member 3 (*EIF4E*) gene, which are essential factors for RNA translation through the control of the polyadenylation tails and the 3′UTR length of EMT- and metastasis-related genes [53]. Similarly, a significant increase was also observed in the expression of the splicing factor 3b subunit 1 (*SF3B1*), which is frequently mutated in Erα+ BCs and is associated with aberrant splicing and a poor prognosis in BC patients [54,55]. ApoERα silencing also induced the expression of QKI, the KH domain containing the RNA binding (*QKI*) gene, which is an RBP that regulates the expression of linear and circular RNA transcripts during EMT in human mammary epithelial cells [25]. *QKI* was also found to correlate with the expression of EMT markers and its high expression was associated with worse overall and disease-free survival times in BC patients [56]. Taken together, our findings confirm the crucial role of apoERα activity in maintaining the luminal epithelial phenotype in BC by promoting the expression of epithelial SF genes and preventing the expression of mesenchymal SF genes.

Notably, apoERα could regulate several SFs acting at the splicing level, as suggested by significant isoform changes in 20 SF genes. Interestingly, the exploration of a recent ERα HITS-CLIP sequencing dataset [19] revealed that genes involved in 227 (27.52%) apoERα-modulated ASEs possess at least one ERα CLIP peak mapped within their gene bodies (Appendix A). Interestingly, among these, 28 ASEs showed an ERα CLIP peak located in the vicinity of the differentially spliced region. Among these ASEs, six concerning events involving SFs (*SAMD4A*, *CELF1*, *QKI*, *ZC3H14*, *SF1*, and *SRSF10*) were correlated with ERα mRNA levels in primary tumors, whose binding motifs were enriched in the apoERα-modulated ASEs (Appendix A). A group of 18 RBPs not included in the RBP-binding motif enrichment analysis were significantly correlated with ERα mRNA levels in primary tumors, as well as exhibiting significant AS changes upon apoERα depletion, and their differentially spliced regions overlapped with ERα CLIP peaks (Appendix A). In particular, *SAMD4A*, which is a conserved RBP across mammals that controls gene translation and stability, has been recently reported as a BC suppressor. Specifically, it destabilizes the expression of proangiogenic transcripts by physically interacting with the stem loop structure in their 3′UTR through its sterile alpha motif (SAM) domain [57]. Moreover, we show here that *SAMD4A* is negatively correlated (r = −0.44, *p* < 0.0001) with ERα mRNA levels in ERα+ BCs, and accordingly, it was reported to be repressed in BC tissues and cancer cell lines where its low expression is associated with the poor survival of patients and its overexpression inhibited tumor angiogenesis and cancer progression [57]. Another EMT regulator RBP, *QKI*, is repressed by apoERα and negatively correlated (r = −0.37; *p* < 0.0001) with ERα mRNA levels in Erα+ BCs, as previously reported [58]. Cao et al. showed that *QKI* suppresses BC progression by binding to the RASA1 transcript and thus increases its mRNA stability, as well as inactivating the MAPK signaling pathway [58]. Noticeably, we took the advantage of our previously published work of an extensive analysis of ERα genomic distribution and regulated genes in MCF-7 cells under different culture conditions [59]. Considering the apoERα chromatin binding sites identified in that study (Appendix A), we found that among the apoERα-regulated RBP/SF genes, 23 showed an apoERα peak in the vicinity of their promoters (Appendix A) and 176 RBPs overlapped with the ERα peak at distal binding sites located 20 kb and 100 kb from the transcription start sites (Appendix A).

Moreover, an isoform switching analysis revealed the different aspects of RNA processing modulated by apoERα, particularly ES and 3′end processing. The functional importance of such mechanisms has been previously reported as a recurrent event involved in cancer development and progression [60], as well as in epithelial BC cells under EMT-inducing treatments [61,62]. Indeed, recent studies showed that 3′UTR length differs among ERα+ and ERα- BC subtypes and that 3′UTR shortening events contribute to tumor growth by interfering with the stability of an endogenous competitive RNA (ceRNA) network in ERα- tumors, especially in association with the aggressive and metastatic phenotypes [63]. In line with previous research, silencing apoERα induced a global 3′UTR lengthening, rather than shortening events [63]. Importantly, genes with a 3′UTR lengthening event were significantly enriched in terms of the regulation of the cellular response to stress, the DNA checkpoint, the positive regulation of the cell cycle, the cell junction organization, and the protein localization to the membrane. A particular example of apoERα-mediated 3′end processing is the isoform switching event in *CELF1* isoforms, which resulted in the upregulation of the isoform with a shorter 3′UTR and the downregulation of the isoform with the longer 3′UTR. Noticeably, although no regulation at the gene level was observed, four *CELF1* isoforms were regulated and responded in opposite directions to apoERα silencing, which explains the overall change at the gene level. Strikingly, the analysis of ChIP-seq data revealed that the spliced-out region of *CELF1* transcript overlapped with the binding of several TFs, including ERα, CTCF, TRIM24, SPDEF, AHR, DNMT3A, RARG, and TP63 (Appendix A), most of which are DE upon apoERα depletion in MCF-7 cells. Moreover, the switching *CELF1* isoforms encoded two protein isoforms that differred in their sequence by a hydrophobic alanine residue at position 104, which overlapped with a splice site [64]. Interestingly, the RBP-binding motif enrichment analysis revealed an enrichment of the *CELF1* binding motif in 320 ASEs, particularly represented by ES events (Appendix A). Moreover, predicted binding sites on 179 apoERα-regulated ASEs overlapped with *CELF1* binding peaks reported by a CLIP-Seq experiment in Hela cells (Appendix A) [65]. The genomic distribution of the predicted *CELF1* binding sites on our list of ASEs revealed that there were about twice as many intronic than exonic bindings of *CELF1*, in line with the genomic distribution of the binding clusters [65]. Thus, apoERα-mediated regulation of *CELF1* at the isoform level could explain, in part, the observed AS changes identified in our dataset. Furthermore, Le Tonquèze et al. identified a high number of *CELF1* binding sites within the 3′UTR regions of the target transcripts, one of the significantly regulated events in our dataset. We identified that 69 genes showing 3′UTR shortening/lengthening events overlapped with the binding sites on 3′UTRs identified by the CLIP-Seq experiment in Hela cells [65], further suggesting that *CELF1* could be involved in the splicing-level events identified in our dataset.

Interestingly, a number of apoERα-modulated ASEs were confirmed to be differentially spliced in BC samples. For instance, the top significant ASE regulated in MCF-7 cells corresponded to the ES of the 7th exon of the amyloid beta precursor-like protein 2 (*APLP2*). The same exon was previously reported by AS-sensitive microarrays to be differentially spliced between MCF-7 and MDA-MB-231, or human mammary epithelial cells (HMEC), showing a higher inclusion level in MCF-7 as compared to other cell lines [66].

The association analysis of the identified apoERα-modulated ASEs with BC clinical outcomes revealed a number of events that were significantly associated with the OS and DFS of ERα+ BC patients. In particular, the ASEs involving exons E3, E4, and E6 in the *COL6A3* gene showed a positive association with patient OS and DFS in ERα+ BC patients. The *COL6A3* gene encodes the α3 chain of the COL6 protein and is formed by a short triple helical (TH) non-collagenous domain of 200 repeating amino acids, 5 C-terminal domains (C1–C5), and 10 (N1-N10) tandem globular N-terminal modules like the von Willebrand factor type A (vWF-A) domain, each encoded by a single exon. The tumor-specific AS of E3, E4, and E6 resulted in the production of protein isoforms either including or lacking the N7, N9, or N10 domains. The expression of the three exons (E3, E4, and E6) was tumor-specific in different cancer types and was associated with the patient’s clinical outcome [67,68,69]. In colorectal cancer, higher inclusion levels of the E5-E6 junction were specifically associated with better OS [68]. In line with these studies, we confirmed an increased inclusion of these three exons in ERα+ BCs and provided further evidence on its association with OS.

## 4. Materials and Methods 

### 4.1. RNA-Seq Read Preprocessing, Alignment, and Expression Quantification 

Raw reads were assessed for Phred quality scores using the FASTQC program (https://www.bioinformatics.babraham.ac.uk/projects/fastqc/, accessed date 1 January 2021), and low bases and adaptor sequences were trimmed off using Fqtrim (http://ccb.jhu.edu/software/fqtrim/, accessed date 1 January 2021) retaining reads of 76 bp length only. Then, clean reads were aligned against the human reference genome (GRCh38.p10) with Gencode v27 annotation (gencode.v27.annotation.gtf.gz) using STAR v2.5.1b [70]. STAR was run in a two-pass mode, allowing an alignment to the transcriptome coordinates by setting the option --quantMode to TranscriptomeSAM. Summary statistics of read alignments are given in Appendix A. The expression levels in read counts, (Transcript per million fragments mapped) TPM, and (Fragments Per Kilobase of exon per Million mapped reads) FPKM units were then estimated at both gene and isoform levels by running RSEM [71] on the alignment files in default parameters.

### 4.2. The Differential Expression Analysis

Differentially expressed genes and isoforms in apoERα silencing, as compared to the control condition, were identified using the DESeq2 R package (v1.26.0) with default parameters [72]. The expression at the isoform level was summarized to the gene level using the *tx-import* bioconductor package [37] and the resulting count matrices were provided to DESeq2. Prior to the DE analysis, genes and isoforms with a low expression were discarded from the analysis and only genes or isoforms with more than 10 normalized read counts in at least one condition (3 out of 6 samples) were considered for further downstream analyses. A gene or isoform was considered as differentially expressed if its associated BH-adjusted *p*-value was < 0.05. All data visualization plots were made using the *ggplot2* R package (v.3.2.1) [73]. Raw and processed RNA-seq data were previously deposited at GSE108693.

### 4.3. The Gene Ontology Enrichment Analysis

The gene ontology terms enriched for up-regulated and down-regulated genes were obtained using the Gene Annotation and Analysis Resource Metascape program [74]. The list of up-regulated and down-regulated genes were analyzed separately and using the Single List Analysis option. The statistically enriched GO terms related to each category of genes were obtained from the GO Biological Processes. Only terms that were associated with an enrichment factor >1.5 and an accumulative hypergeometric test adj. *p*-value < 0.05 were considered significant. To reduce redundancy, the GO terms showing a high number of overlapping genes and a large degree of redundancies were clustered into groups based on their degree of similarities and each group or cluster was represented by the top significant GO term. The top 20 significant clusters were selected for visualization purposes.

### 4.4. The Isoform Switching Analysis

To test for isoform switching events, the IsoformSwitchAnalyzeR tool was applied [75]. Briefly, from the RNA-seq data, the tool takes isoform expression levels quantified in TPM units normalized to transcript length as inputs, and then calculates an isoform fraction (IF) ratio by dividing the isoform expression with the expression of the parent gene (TPM_iso_/TPM_gene_). Expressed genes with less than 1 TPM and expressed isoforms that did not contribute to the expression of the gene (IF < 0.01) were excluded from downstream analyses. The IF was then calculated per each of the remaining isoforms and per condition. For each isoform, a dIF (IF_silencing_−IF_control_) representing the difference in isoform usage between the two conditions was calculated. A cut-off criterion was applied by selecting only those isoforms for which apoERα silencing induced a significant change (BH-corrected *p*-value ≤ 0.05) in IF by at least 10% (i.e., |dIF| > 0.1). Next, the sequences of isoforms showing significant switching events upon apoERα silencing were extracted and annotated for the presence of signal peptide sequences, coding potential, and for their associated pfam protein domains. The biological consequences of the observed switches, including IR, domain gain/loss, coding/non-coding potential, and the shortening/lengthening of the open reading frame were then evaluated for the switching of isoforms from the same parent gene. Next, according to the applied annotation on the switching isoforms, genes were classified into genes with or without downstream functional consequences.

### 4.5. The Differential Alternative Splicing Analysis

The list of differentially regulated AS events upon apoERα silencing were identified using rMATS [76,77]. All the sequences and annotations used in this analysis were based on GRCh38 genome assembly and Gencode v27 annotation. To ensure the quantification of expressed events, a prefiltering criteria was applied by only considering those splicing events whose supporting reads were at least 10 in at least two samples per condition. In addition, a splicing event with a ΔPSI value between the silencing and control conditions of less than 5% (|ΔPSI| < 0.05) or that was associated with adj *p*-value of > 0.05 were excluded from the downstream analysis.

### 4.6. The RBP Binding Motif Enrichment Analysis

To identify RNA-binding proteins as putative regulators of the observed changes in each splicing event identified, the sequences of the regulated ASEs extended ± 200 nucleotides on both sides were scanned for the occurrence of RBP binding motifs. In case of MX events, the regions involved in ASEs were extended on both sides by 100 nucleotides only. The RNA binding motifs for 105 different splicing factors were collected from the RNAcompete study [78]. Next, the MoSEA package was used to search the sequence of the ASEs for the occurrence of RBP binding motifs [29]. The tool FIMO [79] was used to scan the sequences of the ASEs for the presence of the RBP motifs using a *p*-value < 0.001 as a cut-off. The binding motif enrichment was performed by comparing the number of occurrences of the binding motifs of the RBPs in the regulated ASEs with that observed in a pool of 100 randomly selected sequences of the same size from equivalent regions in non-regulated ASEs (|ΔPSI| < 0.01 and *p* > 0.05). Motif enrichment was performed separately for the two directions of splicing changes (ΔPSI > 0.05 or ΔPSI < −0.05). An enrichment z-score per RNA binding motif, region, and direction of regulation was calculated by normalizing the observed frequency in the regulated ASEs set with the mean and standard deviation of the 100 random control sets. The 100 random control sequences were sampled from non-regulated ASEs for each region of regulation. An RBP was considered as enriched if it was associated with a z-score of > 1.96.

### 4.7. An Overlap with Alternative Splicing Events in Primary Tumor Data

The analysis of ASEs in BC tissues was performed considering the annotations from SpliceSeq [80]. This database reports the PSI values of different ASEs detected in the RNA-Seq data from TCGA. Specifically, the analysis was performed by retrieving all the PSI values of TCGA BRCA cohort from the database website (http://projects.insilico.us.com/TCGASpliceSeq/, accessed date 2 April 2021). Then, the ASE coordinates from the database were converted to hg38 assembly using Liftover (https://genome.ucsc.edu/cgi-bin/hgLiftOver, accessed date 2 April 2021) and overlapped with the ASE coordinates from the rMATS analysis. A Spearman correlation analysis was performed to evaluate the relationship between *ESR1* in FPKM and the PSI value associated with each ASE. A Wilcoxon rank-sum test was performed to evaluate the differences in PSI value distributions between data from patients divided by *ESR1* expression levels, by ERα+ and ERα- BCs, or by Erα+ BCs and normal breast tissue. A correlation analysis between the ASE inclusion/exclusion levels and signaling pathways from the MSigDB hallmark gene sets was performed using PEGASAS in default settings [35].

A survival analysis was performed by collecting OS and DFS information of 773 ERα+ samples from TCGA GDC portal [28], together with PSI values of the analyzed ASEs from the TCGASpliceSeq database [34]. Samples with a PSI value greater than the median were classified as a high expression of the ASE, and samples with a PSI value less than this threshold were classified as a low expression of the ASE. The analysis was performed using the survival v3.2.11 R package.

### 4.8. The Correlation Analysis between ERα mRNA Levels and RNA-Binding Proteins Encoding Genes in ERα+ Breast Tumor Samples

The analysis of the correlation between ERα mRNA levels and that of RBP genes in 773 ERα+ samples was performed retrieving the illuminahisep_rnaseqv2-RSEM_genes_normalized dataset and clinical data (BRCA.clin.merged.txt) from the BROAD GDAC Firehose (https://gdac.broadinstitute.org/, accessed date 10 July 2021) database. Prior to calculating the correlation coefficient of the ERα and RBP genes’ mRNA levels, gene read counts were log2 transformed. A GO enrichment analysis was performed separately for correlating and anti-correlating RBPs using metascape [74]. The correlation between ERα and selected RBPs was represented as a scatterplot using the ggplot2 R bioconductor package [73].

### 4.9. The Overlap with ERα HITS-CLIP Data

The overlap with ERα HITS-CLIP data from previous research was performed considering the genomic coordinates of CLIP peaks provided as Appendix A of the manuscript, which was converted from the hg19 to hg38 genome assembly using Liftover [19]. Then, the CLIP peak coordinates were overlapped with the genomic region spanning genes involved in ASEs, or with the region spanning the exons involved. This region was extended by +/− 200 bp as for the RBP motif enrichment analysis.

## 5. Conclusions

In this work, we unravel a novel layer of gene expression regulation mediated by ERα. First, among apoERα-regulated genes, there is a significant number of RNA binding proteins and splicing factors. Second, this was paralleled by significant changes at the level of the alternative splicing of many transcripts. Third, we observed that these changes are not limited to MCF-7 cells, but are also detectable in primary breast tumors, as correlated with Erα mRNA levels. Thus, we foresee that this novel feature should be considered when studying the functional roles of Erα in the onset and progression of BC. To fully decipher the mechanisms in which Erα is directly or indirectly involved, further studies are necessary, especially in the light of the recent discovery that ERα is itself an RBP. How the basal activity of ERα is modulated by hormones, antagonists, and kinase cascades should be addressed by HITS-CLIP and functional studies.

On the other hand, our computational approach is particularly interesting in identifying isoform level changes that are not observed when considering the gene level and provides a way to predict the downstream functional consequences of these changes at the protein isoform level.

## Figures and Tables

**Figure 1 cancers-13-06261-f001:**
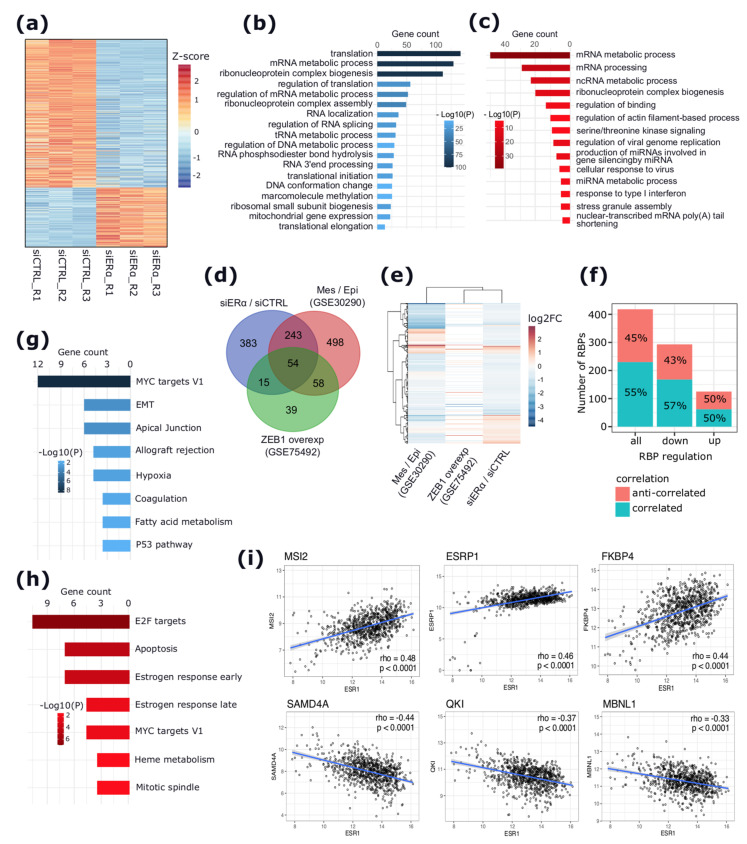
RBPs gene expression changes induced by apoERα in MCF-7 cells and their correlation with ERα mRNA levels in primary tumors. (**a**) Heat map reporting the expression levels of RBPs in siCTRL and siERα conditions ranked by z-score. (**b**,**c**) Bar plots representing significantly enriched processes related to apoERα-induced (**b**) and apoERα-repressed RBPs (**c**)**,** respectively. (**d**) Venn diagram showing the overlap between regulated RBPs in this study and those in GSE30290 and GSE75492 datasets. (**e**) Heat map showing the log2FC of overlapping RBPs between the three studies in (**d**). Missing RBPs from each study are reported in white color. (**f**) Stacked bar plot reporting the number of RBPs positively or negatively correlated with ERα mRNA levels in primary tumors (y-axis) and their regulation by apoERα activity depletion in MCF-7 cells (x-axis). (**g**,**h**) Bar plots reporting enriched gene sets hallmarks related to RBPs that are anti-correlated (**g**) and correlated (**h**) with ERα mRNA levels in primary tumors, respectively. (**i**) Scatter plots reporting six selected RBPs showing significant correlations with ERα mRNA levels in primary tumors.

**Figure 2 cancers-13-06261-f002:**
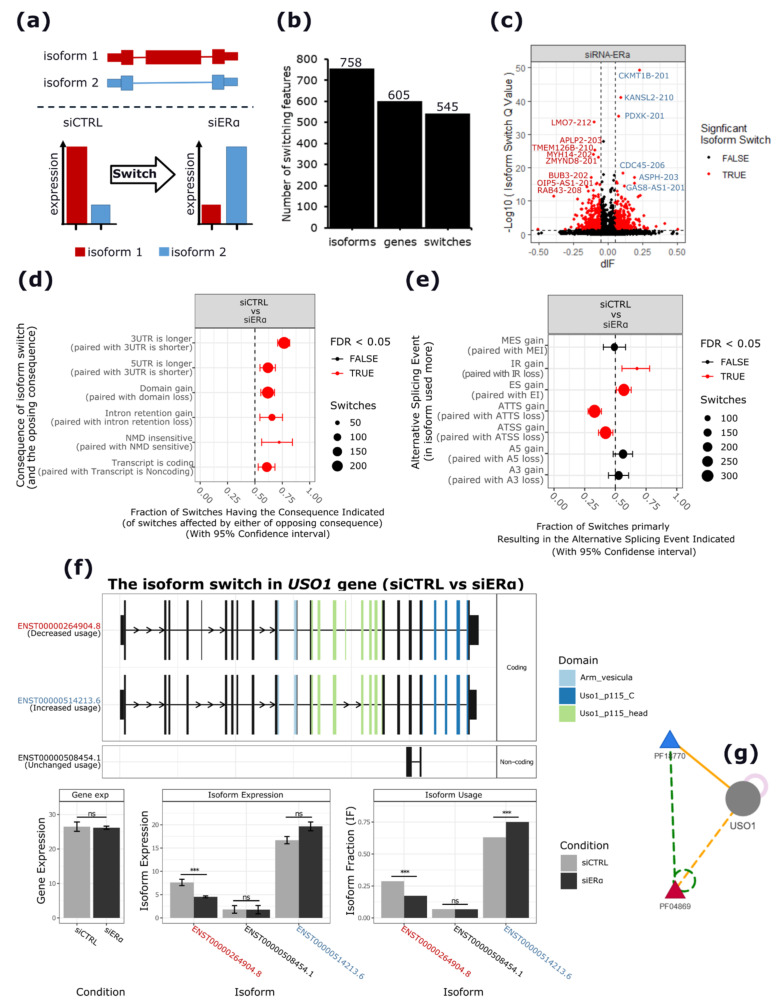
Isoform switching events observed upon apoERα silencing in MCF-7. (**a**) Scheme depicting the principle behind differential isoform usage (dIU) analysis. (**b**) Bar plots reporting the number of significantly switching genes and isoforms involved in switching events. (**c**) Volcano plot reporting the differential isoform fraction (dIF) and relative significance (−log10(adj. p.)) of switching isoform pairs. The top significant switches are labeled accordingly. (**d**) Plot showing the fraction of switches affected by either of the opposing consequences (x-axis) for each pair of opposite consequences (y-axis). Fractions significantly different from 0.5 indicated an enrichment of isoforms with the indicated consequences. (**e**) ASEs enrichment analysis reporting the fraction of switches resulting from each specific ASE, such as MES/MEI, multiple exon skipping/inclusion; ES/EI, exon skipping/inclusion; A5′/A3′, alternative 5′/3′ splice sites; IR, intron retention; ATTS/ATSS, alternative transcription termination/start sites. (**f**) Isoform switching plot for *USO1* gene. Upper panel shows the isoforms involved in the switch and the encoded protein domains. Histograms show the gene and isoform expression levels in normalized TPM units and their DE status (ns, not significant; ***, *p* < 0.0001). (**g**) Downstream effects of *USO1* isoform switching event. Triangles represent the protein domains of the protein (grey circle). The domain encoded by the skipped exon is highlighted in red whereas its interacting domain is in blue. Dashed lines represent the suppressing effects of the ES event.

**Figure 3 cancers-13-06261-f003:**
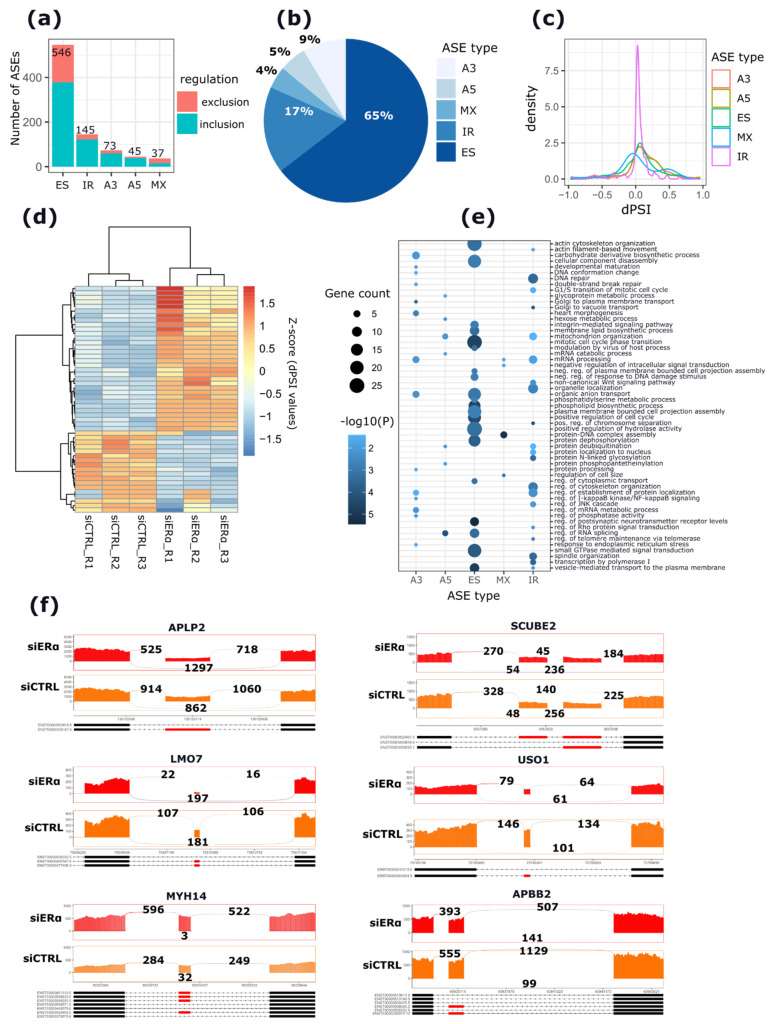
Overview of differential AS changes occurring upon apoERα depletion in MCF-7 cells. (**a**) Stacked bar plot representing the number of significant ASEs for each AS type. Red and cyan colors represent the number of included and repressed ASEs, respectively. (**b**) Pie-chart representing the percentage of ASE types. (**c**) Density plot representing the dPSI of the significant ASEs. (**d**) Heat map reporting the dPSI values of the top 50 significant ASEs. Color bar intensities are proportional to the inclusion level of each event (Z-score). (**e**) Dot plot representing the GO enrichment analysis of genes harboring significant ASEs. The x-axis represents the different ASE types. The size of the dots is proportional to the number of genes enriched in each GO term. The color of the dots is proportional to the significance of the enrichment (−log10 (P)). (**f**) Sashimi plots for the six top significant ASEs. Alternative exons (in red) and their flanking constitutive exons involved in each event are reported. The numbers above junctions indicate the total number of reads supporting either inclusion or exclusion of the ASE.

**Figure 4 cancers-13-06261-f004:**
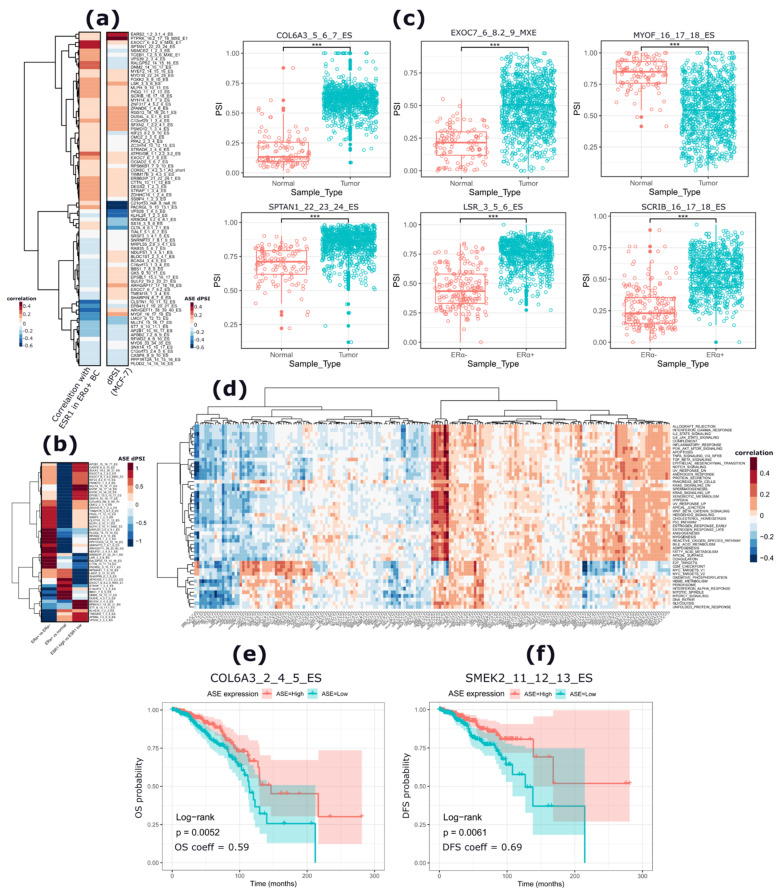
Analysis of the apoERα-regulated ASEs in normal and BC tissues. (**a**) Heat map reporting the dPSI levels in MCF-7 and the correlation coefficient of the ASEs expression (PSI value) with *ESR1* expression in ERα+ BC samples from the BRCA TCGASpliceSeq database. The ASEs are labeled as their gene name, exons involved in the event (regulated and flanking exons), and the type of event. (**b**) Heat map reporting dPSI values of ASEs compared among (i) ERα+ vs. ERα-, (ii) ERα+ vs. normal, and (iii) high *ESR1* vs. low *ESR1* expressing patients. The plot reports 51 ASEs which were significant (*p* < 0.05) in all comparisons. (**c**) Box plots reporting the PSI values of selected ASEs whose inclusion levels are different among compared groups (tumor vs. normal) and (ERα+ vs. ERα-) patients. Wilcoxon *p*-value (***, *p* < 0.00001). (**d**) Heat map representing the correlation of ASE PSI values of ERα-regulated ASEs with enriched hallmarks as reported using PEGASAS algorithm [35]. (**e**,**f**) Survival analysis plots showing the top significant ASE associated with overall survival (OS) (**e**) and disease-free (DFS) survival (**f**) of Erα+ BC patients. Patients are divided into a high expression (ASE = high) and a low expression event (ASE = low) based on the median PSI calculated among all patients.

## Data Availability

Analyzed data are deposited in Gene Expression Omnibus with the identifier GSE108693.

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
