# Peer review of "The Estrogen Receptor α Signaling Pathway Controls Alternative Splicing in the Absence of Ligands in Breast Cancer Cells"

_cancers, 2021, doi:10.3390/cancers13246261_

Round 1

Reviewer 1 Report

The authors applied RNA-seq dataset of MCF-7 cells after siRNA-induced ERα downregulation from database to analyze transcription and alternative splicing. They used various bioinformatical tools to show that RBPs, splicing, are important regulatory factors by which apoERα control EMT. They further showed that the inclusion levels of alternatively spliced exons are associated with overall and disease-free survival. Bioinformatical analysis is clear and strong. The manuscript is acceptable for “cancers” with following modifications.

  1. figure 1, expression changes in RBPs of ERα siRNA was analyzed. Is there clip-seq data available in MCF7 versas primary breast cells? The comparison of clip data in these two cells should be able to provide the importance of RBPs in ERα regulation.
  2. Figure 3f, sashimi plots are shown to demonstrate the alterations in alternative splicing. However, experimental evidences (RT-PCR) are required to validate the bioinformatical analysis.

Author Response

Reviewer: The authors applied RNA-seq dataset of MCF-7 cells after siRNA-induced ERα downregulation from the database to analyze transcription and alternative splicing. They used various bioinformatical tools to show that RBPs, splicing, are important regulatory factors by which apoERα controls EMT. They further showed that the inclusion levels of alternatively spliced exons are associated with overall and disease-free survival. Bioinformatical analysis is clear and strong. The manuscript is acceptable for “cancers'' with following modifications.

  1. figure 1, expression changes in RBPs of ERα siRNA was analyzed. Is there clip-seq data available in MCF7 versus primary breast cells? The comparison of clip data in these two cells should be able to provide the importance of RBPs in ERα regulation.

Authors: We thank the Reviewer for his comment. We carefully checked out publicly available CLIP-Seq datasets at different databases, such as POSTAR3 database (http://postar.ncrnalab.org/), STARbase (https://starbase.sysu.edu.cn/), and ENCODE, and found that, in most cases, these experiments are performed in cell lines different from MCF-7 cells. Then, to further investigate the relationship between ERα signaling and RBPs expression in breast cancer, we performed an additional expression analysis of RBPs in breast cancer cell lines and primary tissues. Specifically, to better understand whether ERα-regulated RBPs are functionally relevant in primary tumor tissues, we performed an overlap between the list of identified apoERα-regulated RBPs and those reported as differentially expressed in an ERα+ tumors versus normal adjacent tissues comparison, reported by [1]. This overlap indicates that 413 (out of 681; 60.73%) RBPs are differentially expressed in both datasets, most of which (238, 56%) are upregulated in tumors (versus normal adjacent tissues) while they are downregulated upon apoERα depletion in MCF-7 cells. Furthermore, we also found a previously published study [2] where authors performed a gene co-expression analysis of RBP/SF genes in different breast cancer subtypes and their correlation with clinical phenotypes (aggressiveness, metastasis) of these tumors was investigated. Particularly, the authors in this study show the association between the expression of SFs and ERα positivity status. Specifically, they identified 24 RBPs significantly related to ERα expression and among them 21 were included in our set of apoERα-regulated RBPs. These results further support the relationship between the regulatory activity of ERα on the expression of these genes. Authors also reported 245 genes as DE comparing tumor to normal samples or metastatic to primary breast tumors, respectively, among which 76 were included in our analysis and whose expression was modulated by apoERα. These results were summarized in Supplementary Table 10 and added to the discussion section of the revised version of the manuscript.

Reviewer:

  1. Figure 3f, sashimi plots are shown to demonstrate the alterations in alternative splicing. However, experimental evidences (RT-PCR) are required to validate the bioinformatical analysis.

Authors: The problem of validating alternative splicing events detected by RNAseq is well present and largely debated in the AS community. Variations in alternative exon inclusion/exclusion are usually very small. This is also true in our case, by looking at the distribution of dPSI values (Figure 3c) of the significant events you will see that they are all between -0.20 and 0.2 (20%), making it difficult to observe differences in the bands on the gel. In addition, please consider that, technically, such a narrow range of fold-change is definitely out of the range of reliable detection by either end-point and real-time PCR techniques. In addition, other issues can hinder the validation of AS events, including the presence of very small AS exons (< 30 nt) as in our dataset, the presence of unannotated transcript isoforms in the reference transcriptome, and the possible presence of genomic variants (e.g., indels) that may significantly modify the amplicon size. This is particularly relevant in cancer cell lines where a high heterogeneity of transcript variants was previously reported [3–7]. All these factors should be considered for a successful experimental validation of alternative splicing events. We have indeed attempted validation of several AS events, both by real-time RT-PCR and by end-point RT-PCR followed by gel electrophoresis, with variable success. As an example, we focused on the AS event involving USO1 gene, which is shown within the manuscript in Figure 2f. This exon skipping event involved the 15th exon (ENSE00001735719) of isoform ENST00000264904 that is less included upon apoERα silencing. As reported in the attached figure, a decrease of the inclusion form was confirmed by PCR and amplicon separation by 10% polyacrylamide  electrophoresis gel.
Furthermore, we kindly invite you to consider the fact that, as clearly reported within our manuscript, a significant proportion of the predicted splicing events were also observed and found to be differentially regulated in primary breast tumor samples (See Results, section 2.5, Figure 3 and Supplementary Table 7), confirming that they are not anecdotic of our dataset.

Figure legend: Experimental validation of USO1 exon skipping by RT-PCR. (a) Isoform switching plot showing increased (ENST000000264904.8) and depleted (ENST000000514213.6) gene isoforms upon apoERα depletion. The switching isoforms pair differ by differential exclusion of the alternative exons 7 and 14. (b) Sashimi plot reporting the differential inclusion of the exon 14 highlighted with black rectangle in (a). (c) Experimental validation of the exon skipping event (middle black exon), confirming the results of the bioinformatic analysis. The amplicons of 167 and 146 base pairs were obtained using a pair of primers (forward: 5’-tgctcagggttcaacttgct-3’; reverse: 5’-gggacaattgcttagccagg-3’) which target both inclusion and exclusion isoforms involved in the switching event. PCR products were resolved on a 10% polyacrylamide electrophoresis gel.

References
1.     Sebestyén, E.; Singh, B.; Miñana, B.; Pagès, A.; Mateo, F.; Pujana, M.A.; Valcárcel, J.; Eyras, E. Large-Scale Analysis of Genome and Transcriptome Alterations in Multiple Tumors Unveils Novel Cancer-Relevant Splicing Networks. Genome Res. 2016, 26, 732–744.
2.     Koedoot, E.; Smid, M.; Foekens, J.A.; Martens, J.W.M.; Le Dévédec, S.E.; van de Water, B. Co-Regulated Gene Expression of Splicing Factors as Drivers of Cancer Progression. Sci. Rep. 2019, 9, 5484.
3.     Vincent, K.M.; Findlay, S.D.; Postovit, L.M. Assessing Breast Cancer Cell Lines as Tumour Models by Comparison of mRNA Expression Profiles. Breast Cancer Res. 2015, 17, 114.
4.     Namba, S.; Ueno, T.; Kojima, S.; Kobayashi, K.; Kawase, K.; Tanaka, Y.; Inoue, S.; Kishigami, F.; Kawashima, S.; Maeda, N.; et al. Transcript-Targeted Analysis Reveals Isoform Alterations and Double-Hop Fusions in Breast Cancer. Commun Biol 2021, 4, 1320.
5.     Yu, K.; Chen, B.; Aran, D.; Charalel, J.; Yau, C.; Wolf, D.M.; van ’t Veer, L.J.; Butte, A.J.; Goldstein, T.; Sirota, M. Comprehensive Transcriptomic Analysis of Cell Lines as Models of Primary Tumors across 22 Tumor Types. Nat. Commun. 2019, 10, 3574.
6.     Kleensang, A.; Vantangoli, M.M.; Odwin-DaCosta, S.; Andersen, M.E.; Boekelheide, K.; Bouhifd, M.; Fornace, A.J., Jr; Li, H.-H.; Livi, C.B.; Madnick, S.; et al. Genetic Variability in a Frozen Batch of MCF-7 Cells Invisible in Routine Authentication Affecting Cell Function. Sci. Rep. 2016, 6, 28994.
7.     Anvar, S.Y.; Allard, G.; Tseng, E.; Sheynkman, G.M.; de Klerk, E.; Vermaat, M.; Yin, R.H.; Johansson, H.E.; Ariyurek, Y.; den Dunnen, J.T.; et al. Full-Length mRNA Sequencing Uncovers a Widespread Coupling between Transcription Initiation and mRNA Processing. Genome Biol. 2018, 19, 46.

Reviewer 2 Report

cancers-1462728

The Estrogen Receptor α Signaling Pathway Controls Alternative Splicing In Absence Of Ligand In Breast Cancer Cells

Jamal Elhasnaoui, Giulio Ferrero, Valentina Miano, Santina Cutrupi and Michele De Bortoli

Elhasnaoui et al. investigated the role of estrogen receptor (ER) α in regulating the expression of RNA-binding proteins in luminal breast cancer cells. They tried to determine the effects of ERα downregulation at the isoform level by exploring changes in isoform usage and alternative splicing. They found a novel layer of gene regulation mediated by ERα in cell growth and EMT.

The authors successfully identified many interesting changes with ERα knockdown. The data presented here are very interesting and attracting many RNA biology scientists.

I have a few comments and suggestions as follows.

1) Although the authors tried knockdown of ERα, the data they successfully downregulated it are not shown. Immunoblotting experiments for ERα should be done and shown in the Figures.

2) It would be nicer if he authors could demonstrate isoform switch of some genes by RT-PCR with gel electrophoresis in order to validate their results.

3) It would also be wonderful if the authors could discuss promoters of the genes affected by ERα knockdown.

Author Response

Reviewer: Elhasnaoui et al. investigated the role of estrogen receptor (ER) α in regulating the expression of RNA-binding proteins in luminal breast cancer cells. They tried to determine the effects of ERα downregulation at the isoform level by exploring changes in isoform usage and alternative splicing. They found a novel layer of gene regulation mediated by ERα in cell growth and EMT.

The authors successfully identified many interesting changes with ERα knockdown. The data presented here are very interesting and attract many RNA biology scientists.

I have a few comments and suggestions as follows.

Reviewer: 1) Although the authors tried a knockdown of ERα, the data they successfully downregulated it are not shown. Immunoblotting experiments for ERα should be done and shown in the Figures.

Authors: We thank the Reviewer for his comment. We have previously shown that ERα is successfully downregulated in the same samples used to generate the RNA-seq data analyzed in this study. This result has already been published in our previous paper (Miano, et al., 2018) [1] in the Supplementary Figure (1d).

Reviewer: 2) It would be nicer if the authors could demonstrate isoform switch of some genes by RT-PCR with gel electrophoresis in order to validate their results.

Authors: The problem of validating alternative splicing events detected by RNA-seq is well present and largely debated in the AS community. Variations in alternative exon inclusion/exclusion are usually very small. This is also true in our case, by looking at the distribution of dPSI values (Figure 3c) of the significant events you will see that they are all between -0.20 and 0.2 (20%), making it difficult to observe differences in the bands on the gel. In addition, please consider that, technically, such a narrow range of fold-change is definitely out of the range of reliable detection by either end-point and real-time PCR techniques. In addition, other issues can hinder the validation of AS events, including the presence of very small AS exons (< 30 nt) as in our dataset, the presence of unannotated transcript isoforms in the reference transcriptome, and the possible presence of genomic variants (e.g., indels) that may significantly modify the amplicon size. This is particularly relevant in cancer cell lines where a high heterogeneity of transcript variants was previously reported [2–6]. All these factors should be considered for a successful experimental validation of alternative splicing events. We have indeed attempted validation of AS events, both by real-time RT-PCR and by end-point RT-PCR followed by gel electrophoresis. Specifically, we focused our analysis of the AS event involving USO1 gene that is shown within the manuscript in Figure 2f. This exon skipping event involved the 15th (ENSE00001735719) exon of isoform ENST00000264904 that is less included upon apoERα silencing. As reported in the attached figure, a decrease of the inclusion form was confirmed by PCR and amplicon separation by 10% polyacrylamide  electrophoresis gel.
Furthermore, we kindly invite you to consider the fact that, as clearly reported within our manuscript, a significant proportion of the predicted splicing events were also observed and found to be differentially regulated in primary breast tumor samples (See Results, section 2.5, Figure 3 and Supplementary Table 7), confirming that they are not anecdotic of our dataset. 

Reviewer: 3) It would also be wonderful if the authors could discuss promoters of the genes affected by ERα knockdown.

Authors: We thank the Reviewer for his comment. Our group has extensively investigated chromatin interactions of ERα signaling pathway in breast cancer by performing an extensive analysis of multiple omics datasets. Indeed, we previously published a comprehensive analysis of ERα genomic distribution and regulated genes in MCF-7 under different culture conditions (hormone-deprivation, full medium growth, and uponE2 treatment) representing ligand-independent (apoERα), constitutive, and ligand-dependent activities of ERα, respectively [7]. With the respect of these data, we explored the promoters of regulated RBPs identified in our datasets and we confirmed 23 RBP genes (100 isoforms) characterized by proximal apoERα binding sites (within 2,5kb from the gene/transcript TSS) and 176 (100 and 76 within 20kb; 100kb, respectively) with distal binding sites. These results are in line with different evidence for an ERα regulatory activity driven mainly by binding at enhancer sequences, long-range chromatin interactions, and the regulation of a network of transcription factors [8]. We added a comment on this analysis in the novel Discussion section of the manuscript.

References

1.     Miano, V.; Ferrero, G.; Rosti, V.; Manitta, E.; Elhasnaoui, J.; Basile, G.; De Bortoli, M. Luminal lncRNAs Regulation by ERα-Controlled Enhancers in a Ligand-Independent Manner in Breast Cancer Cells. International Journal of Molecular Sciences 2018, 19, 593.
2.    Vincent, K.M.; Findlay, S.D.; Postovit, L.M. Assessing Breast Cancer Cell Lines as Tumour Models by Comparison of mRNA Expression Profiles. Breast Cancer Res. 2015, 17, 114.
3.     Namba, S.; Ueno, T.; Kojima, S.; Kobayashi, K.; Kawase, K.; Tanaka, Y.; Inoue, S.; Kishigami, F.; Kawashima, S.; Maeda, N.; et al. Transcript-Targeted Analysis Reveals Isoform Alterations and Double-Hop Fusions in Breast Cancer. Commun Biol 2021, 4, 1320.
4.     Yu, K.; Chen, B.; Aran, D.; Charalel, J.; Yau, C.; Wolf, D.M.; van ’t Veer, L.J.; Butte, A.J.; Goldstein, T.; Sirota, M. Comprehensive Transcriptomic Analysis of Cell Lines as Models of Primary Tumors across 22 Tumor Types. Nat. Commun. 2019, 10, 3574.
5.     Kleensang, A.; Vantangoli, M.M.; Odwin-DaCosta, S.; Andersen, M.E.; Boekelheide, K.; Bouhifd, M.; Fornace, A.J., Jr; Li, H.-H.; Livi, C.B.; Madnick, S.; et al. Genetic Variability in a Frozen Batch of MCF-7 Cells Invisible in Routine Authentication Affecting Cell Function. Sci. Rep. 2016, 6, 28994.
6.     Anvar, S.Y.; Allard, G.; Tseng, E.; Sheynkman, G.M.; de Klerk, E.; Vermaat, M.; Yin, R.H.; Johansson, H.E.; Ariyurek, Y.; den Dunnen, J.T.; et al. Full-Length mRNA Sequencing Uncovers a Widespread Coupling between Transcription Initiation and mRNA Processing. Genome Biol. 2018, 19, 46.
7.     Ferrero, G.; Miano, V.; Beccuti, M.; Balbo, G.; De Bortoli, M.; Cordero, F. Dissecting the Genomic Activity of a Transcriptional Regulator by the Integrative Analysis of Omics Data. Sci. Rep. 2017, 7, 8564.
8.     Farcas, A.M.; Nagarajan, S.; Cosulich, S.; Carroll, J.S. Genome-Wide Estrogen Receptor Activity in Breast Cancer. Endocrinology 2021, 162, doi:10.1210/endocr/bqaa224.

Round 2

Reviewer 1 Report

The authors addressed all of my points well.